# Consumers’ Willingness to Pay for Imported Milk: Based on Shanghai, China

**DOI:** 10.3390/ijerph17010244

**Published:** 2019-12-29

**Authors:** Lingling Xu, Xixi Yang, Linhai Wu

**Affiliations:** Institute for Food Safety Risk Management, School of Business, Jiangnan University, Wuxi 214122, China; 8383800028@jiangnan.edu.cn (L.X.); 6180906013@stu.jiangnan.edu.cn (X.Y.)

**Keywords:** choice experiment, preference, country of origin, nutrition claim

## Abstract

Against the backdrop of the continuous large-scale growth of imported milk in China, in this research 310 consumers in Shanghai were used as a sample, and a choice experiment was conducted to study consumer preference and willingness to pay for imported milk. The following product attributes were included: nutrition claim, fat content, flavor, country of origin, and price. Our results show that, excepting price, consumers consider flavor the most important attribute, followed by nutrition claim, fat content, and country of origin. Consumers can be delineated into four segments based on consumer preference for the attributes of imported milk: “nutrition claim seekers” are willing to pay the highest price for imported milk with nutrition claims, “indifferent” consumers pay little attention to imported milk attributes, “flavor-oriented” consumers have a strong preference for strawberry-flavored imported milk, and “price-sensitive” consumers weigh the price when choosing imported milk.

## 1. Introduction

The Chinese market has gradually opened up and the tariff on imported milk has decreased from 25% to 15% since China entered the WTO (World Trade Organization). Furthermore, online shopping has eliminated many middleman transactions, greatly reducing the cost of circulation. Currently, most imported milk in China is sold through online shopping platforms, further reducing the circulation cost [1,2]. At the same time, food safety incidents, including "melamine," "scophthalmus maximus," "beta-adrenergic agonist," and "Sudan," have presented huge challenges to consumers’ psychological tolerance and confidence in food [3], and Chinese consumers have begun to favor imported dairy products because of the melamine milk powder incident in 2008 [4]. Mintel’s (2018) report on the milk consumption trend highlighted that Chinese consumers’ attitudes toward local dairy products are clearly differentiated: 44% believe that local milk products are reliable, and 36% do not. Compared with domestic dairy products (34%), more consumers preferred imported dairy products (43%) (The data came from a survey conducted by Mintel in November 2017 among 3000 consumers aged between 29 and 49 in first to third-tier Chinese cities.) [5]. Because of the decreasing costs and increasing consumer preferences, imported milk in China has increased 89.19 times from more than 7000 tons in 2008 to 634,100 tons in 2016. In 2018, China imported 673,300 tons of milk, of which 344,300 tons were from the European Union; 233,000 tons from New Zealand; 81,100 tons from Australia; and the remaining milk was from Denmark, the Netherlands, and so on. Imported milk, high-end milk, and normal-temperature yogurt have become the three main categories dominating the liquid milk market in China. The total sales of these three categories accounted for 70.1% in 2018 [6].

As China’s economy develops and consumer quality improves, consumers are paying increasing attention to the high quality, high nutrition, and low calories of food. Chinese consumers now increasingly prefer foods based on the concepts of healthy eating, balanced nutrition, improved immunity, and reduced risk of disease, such as imported foods; organic foods; foods supplemented with vitamins or calcium, iron, and zinc; low-fat and low-salt foods; and whole-grain foods [7,8,9]. Driven by market demand, China’s imported milk will have a broad market prospect. Therefore, understanding Chinese consumers’ preference for imported milk with different healthy and nutritional attributes will help both domestic and overseas producers accurately grasp the market characteristics of imported milk and better meet consumers’ needs.

Country of origin usually refers to the country where goods or products are processed, manufactured, planted, and grown. Generally, country of origin is considered as a natural label of the quality and characteristics of agricultural products [10]. For consumers, information on geographic origin serves to identify the product and assess its quality [11,12,13]. Some researchers studied consumers’ preferences for domestic and imported food in Japan, Finland, Italy, South Korea, Germany, the United States, etc. The results show that consumers in developed countries prefer domestic food to imported food [14,15,16,17,18,19,20,21,22,23,24,25,26,27,28]. While Nuttavuthisit and Thøgersen [29] and Thøgersen et al. [30] noted that compared with consumers in developed countries, consumers in less developed countries prefer imported food produced in developed countries. Yin et al. [31] and Yin et al. [32] found that compared with domestic infant formula, Chinese consumers prefer infant formula produced in the United States, the European Union, and New Zealand. Chinese consumers’ preference for imported foods differs from that evident in developed countries because of China’s special national conditions and frequent food safety incidents, as Chinese consumers prefer foods from developed countries to domestic foods. However, very few studies focus on the preferences of consumers in developing or less developed countries regarding importing foods from different countries. Furthermore, literature systematically studying Chinese consumers’ preference for imported milk based on different attributes such as nutrition claims, flavors, calories, and country of origin is scant. This study focused on Chinese consumers’ preference and willingness to pay (WTP) for imported milk based on different attributes to contribute to helping domestic and foreign milk producers make decisions.

## 2. Methods

### 2.1. Theoretical Framework and Empirical Model

According to Lancaster’s consumer demand theory and random utility theory [33], different imported milk attributes determine consumers’ preference for imported milk. Based on their own budget constraints, consumers will choose imported milk with different combinations of attributes to maximize their own utility. Suppose $U_{\mathrm{mjk}}$ represents the utility obtained by consumer n selecting the j-th imported milk in the selection task C in the k-th situation. A deterministic part $V_{\mathrm{mjk}}$ and a random part $\varepsilon_{\mathrm{mjk}}$ constitute the utility of the consumer:(1)$$U_{\mathrm{mjk}}=V_{\mathrm{mjk}}+\varepsilon_{\mathrm{mjk}}$$

Only when $U_{\mathrm{mjk}}>U_{\mathrm{mik}}$ or $V_{\mathrm{mjk}}-V_{\mathrm{mik}}>\varepsilon_{\mathrm{mik}}-\varepsilon_{\mathrm{mjk}}$ (∀i ≠ j), individual m will choose the j-th imported milk. Therefore, the probability of individual m selecting the j-th alternative is given by:(2)$$P_{\mathrm{mjk}}=prob(V_{\mathrm{mjk}}-V_{\mathrm{mik}}>\varepsilon_{\mathrm{mik}}-\varepsilon_{\mathrm{mjk}};\forall i\neq j)$$

$V_{\mathrm{mjk}}$ represents a linear function of the five attributes of imported milk, including price, flavor, fat content, country of origin, and nutrition claims:(3)$$V_{\mathrm{mjk}}={\alpha^{\prime }}_{m\;}X_{\mathrm{mjk}}$$
where ${\alpha^{\prime }}_{m}$ is a vector of random parameters indicating the individual preference. $X_{\mathrm{mjk}}$ is the vector of attributes in the j-th alternative.

The probability that the consumer m chooses the j-th imported milk under situation k can be expressed as [34]:(4)$$P_{\mathrm{mjk}}=\int \frac{\exp(V_{\mathrm{mjk}})}{|\sum_{i}\exp(V_{\mathrm{mik}})}f(\alpha_{n})d\alpha_{n}$$
where $f(\alpha_{n})$ represents the probability density function of parameter $\alpha_{n}$, the distribution of $\alpha_{n}$ is specified by $f(\alpha_{n})$ [35]. In the lantent class logit (LCL) model, the utility of consumer m choosing the j-th alternative under situation k is:(5)$$U_{\mathrm{mjk}}{|}_{S}=\alpha_{S}X_{\mathrm{mjk}}+\varepsilon_{\mathrm{mjk}}{|}_{S}$$
where $\alpha_{S}$ is the parameter vector of the segment s related to the explanatory variable, $\varepsilon_{nit}{|}_{S}$ represents an error term.

Therefore, on the premise of s categories, the probability of consumer m choosing option j can be expressed as:(6)$$G_{\mathrm{mj}}=\sum_{s=1}^{S}G_{\mathrm{ms}}\prod_{k=1}^{K}G_{\mathrm{mjk}}{|}_{S}$$
where $G_{\mathrm{ms}}$ is the probability of individual m falling into category s, $G_{\mathrm{mjk}}{|}_{S}$ represents the probability that the consumer m chooses option j in the k-th occasion on the premise of s categories [36].

The attributes and levels of imported milk were all effect codes except price, and the price was a continuous variable. “Chooseno” was a dummy code, its value is 1 when participant chose a non-buying option, otherwise it is 0.

The average WTP of consumers for attribute levels of imported milk can be calculated by the following formula: [34,37]:(7)$${\mathrm{WTP}}_{x}=-2(\frac{\alpha_{x}}{\alpha_{p}})$$
where ${\mathrm{WTP}}_{x}$ is the WTP for imported milk attribute x, $\alpha_{x}$ is the estimated coefficient for attribute x, and $\alpha_{p}$ is the estimated parameter of price.

### 2.2. Experiment Design

This paper studies the consumer’s preference for different properties of imported milk based on a choice experiment. Participants need to make their choices among a group of products or options on the basis of the principle of utility maximization in a selection scenario [38]. The differences in products and the utility of the product are determined by their attributes. A complete choice experiment usually requires the participants to make multiple choices in multiple selection scenarios. According to the decision information of the participants, their preference for each attribute can be analyzed and the value of the attribute can be further evaluated.

#### 2.2.1. Attribute and Level Settings

The attributes and levels were determined based on actual sales of imported milk in China, and the results of related research [39]. Ultimately, five attributes, namely, fat content, flavor, nutrition claims, country of origin, and price, were employed to study Chinese consumers WTP for imported milk. Table 1 shows the imported milk attributes and corresponding design levels.

With the improvement of living standards, an increasing number of people are becoming obese. In many countries, reducing sugar and fat intake in daily diet has not only become a major public health priority, but also has been incorporated into national dietary guidelines [40]. The mass media and public health campaigns also commonly spread that high sugar and fat intake is not good for health [41]. Therefore, consumers are paying increasing attention to information such as fat content, calories, and sugar content in foods, especially female consumers and people who are overweight [42]. Investigating consumer preferences for fat content in dairy foods like milk, Tuorila [43] delineated three levels: skim, 1.9% milkfat, and 3.9% milkfat. Chapman and Lawless [44] used two levels: skim and 2% milkfat; Harwood and Drake [45] employed four levels: skim, 1% milkfat, 2% milkfat, and whole milk; Yasmine et al. [46] and Getter et al. [47] used three levels: skim, reduced milkfat, and whole milk. Considering the limitation of the number of attribute levels and actual sales situation of imported milk in China, this study delineated the fat content of imported milk into two levels: whole milk and skim milk.

Results reported in previous studies confirmed that the flavor of dairy products had a great impact on consumers’ purchase and consumption of various dairy products [46,47,48,49,50]. Currently, most imported milk sold in the Chinese market is plain milk. The proportion of other flavors such as strawberry, banana, and chocolate is very small. For example, in Jingdong Mall, the proportion of plain, strawberry-flavored, banana-flavored, and chocolate-flavored imported milk is 76.53%, 8.42%, 4.08%, and 3.32%, respectively. To further explore consumers’ preference for the flavor of imported milk, this study delineated the flavor attribute thereof into three levels: strawberry flavor, banana flavor, and plain.

Calcium has multiple functions, including vascular contraction and vasodilation, muscle function, nerve transmission, announced by the National Institutes of Health. Only one percent of calcium in human body is needed in the circulation to support these functions, and the remaining ninety-nine percent of calcium is stored in bones and teeth to strengthen and support their structure [51]. Iron is essential for the human body, playing a key role in many metabolic processes, such as oxygen transport, DNA synthesis, energy metabolism, and cell growth and differentiation [52]. Zinc is necessary for protein production, growth, cell division, immune function, DNA synthesis, wound healing, bone and teeth formation, brain activity, skin renewal, and the proper functioning of the nervous system [53]. Vitamin D is indispensable to the human body and the lack of it causes growth retardation and rickets in children, precipitates and exacerbates osteopenia and osteoporosis, and increases the risk of fractures in adults [54,55]. Vitamin A and zinc work together to prevent mouth ulcers, dry flaky skin, poor night vision, and frequent colds. Currently, the imported milk sold in China generally contains a certain vitamin (such as vitamin A, vitamin D, etc.) or element needed by humans (such as calcium, iron, etc.). A small amount of imported milk simultaneously contains calcium, iron, zinc, and vitamin D. Because vitamin A and vitamin D can work together to promote bone development and improve body immunity, we classified nutrition claims into three categories: (a) claim “contains vitamin A, vitamin D”; (b) claim “contains Ca, Fe, Zn, vitamin D”; and (c) no claim.

In the Chinese market, milk imports totaled 673,300 tons in 2018, of which 344,300 tons (51.1%) were from the European Union and 233,000 tons were from New Zealand (34.6%); the total of the two accounted for 85.7%. The remaining milk was from Australia, Denmark, the Netherlands, and so on. Moreover, milk imported from the European Union mostly came from Germany (26%) and France (10%). Due to the limitation of attribute levels, this study delineated the country of origin attribute into three levels: New Zealand, Germany, and France.

The price of imported milk in the Chinese market is different due to the differences in the country of origin, brand, and other attributes. An investigation of the price of 200 mL/250 mL imported milk in Jingdong Mall revealed that the price of regular imported milk without additional nutrients is generally around 4.5 yuan, and the price of imported milk containing one to two types of enhanced nutrients is between 5–7 yuan. The price of organic imported milk and imported milk for children and pregnant women is relatively high, generally between 7–9 yuan. Based on the existing literature and actual Chinese market, the price attribute of imported milk (1 × 200 mL) was delineated into four levels: 4.5 yuan, 5.8 yuan, 7.1 yuan, and 8.4 yuan.

#### 2.2.2. Questionnaire Design

The questionnaire included the following aspects: (i) socio-demographic and economic characteristics, (ii) consumer attitudes toward healthy eating, (iii) objective nutritional knowledge and use of nutrition claims, and (iv) the choice experiment. In the choice experiment, 2 × 3 × 3 × 3 × 4 = 216 virtual choice profiles were generated in terms of the attributes and corresponding levels. Moreover, we used two choice profiles of imported milk, plus a “Neither” profile, totaling three alternative choices for each choice task. Consumers need to compare (216 × 215)/2 = 23,220 groups of choice profiles. According to Rossi et al. [56], fatigue occurs if participants are required to identify more than 15–20 choice profiles. Therefore, for reducing biases and estimating all the cross-terms, determining the number of choice sets and randomly designing the imported milk attribute combinations, it is essential to use the fractional factorial design (FFD). To alleviate consumer fatigue, the number of choice tasks in each questionnaire was finally reduced to 11. Then, we used SSI Web 7.0 to design 12 versions of the questionnaire to ensure that the questionnaire had the highest design efficiency. The test results of the choice task design showed that all attributes’ value of D-efficiency was over 91.71%. In the experimental design, the frequency of all levels and attributes was generally well balanced, and the bias between actual and ideal standard deviation was lower than 1 percent. A “Neither” option, which was adopted from Adamowicz et al.’s research [57], was included to ensure the completeness of each choice task. Figure 1 shows a sample of the choice task card. All subjects gave their informed consent for inclusion before they participated in the study. The study was conducted in accordance with the Declaration of Helsinki, and the protocol was approved by the Ethics Committee of Jiangnan University ([2018]21).

### 2.3. Experimental Organization and Implementation

The survey was conducted in Shanghai, China’s economic and financial center. From the aspects of commercial resource agglomeration, logistics and transportation convenience, and urban people’s activity and lifestyle diversity, Shanghai is far ahead of other cities in China and a “super first-tier city.” People in Shanghai will be more exposed to purchasing imported milk. Therefore, investigating the preferences of consumers in Shanghai for imported milk is representative. In this study, data were collected in seven districts of Shanghai (Huangpu, Xuhui, Jiading, Yangpu, Hongkou, Fengxian, and Songjiang districts). Trained surveyors conducted each direct, random face-to-face interview with the third consumer entering their sight in large plazas integrating catering and entertainment. 

In our survey, respondents were invited to participate only when they were responsible for half or more of their household’s food purchases. Two groups of participants were investigated in each district, with 23–26 participants in each group and 343 participants in seven districts. There were twelve versions of questionnaire, with a total of fourteen groups of participants. Participants from the same group completed the same version of the questionnaire. The remaining two groups of participants randomly chose two versions of the questionnaire to answer. Finally, all 264 (2 × 11 × 12) choice profiles of imported milk generated from the questionnaire were included in the experiment. All participants were informed that the choice task cards of imported milk were different in fat content, flavor, nutrition claims, country of origin and price, and differed in the brand. Finally, 310 valid questionnaires were recovered from 343 questionnaires. All surveys were completed between August 15–22, 2018.

## 3. Results

### 3.1. Description of Sample

As shown in Table 2, for all samples the majority of the 310 respondents were women (56.1%), which reflect that women are responsible for food purchasing in most Chinese households. Overall, 21% of the respondents were aged less than 24 years, 54.8% 25–34 years, 16.8% 35–44 years, and 7.4% more than 45 years. Moreover, 27.1% had a junior college education and 31% had an undergraduate-level education. Regarding annual income, 24.5%, 14.8%, and 7.4% of respondents’ annual household income ranged from 101,000 to 150,000 RMB, 151,000 to 200,000 RMB, and 201,000 to 300,000 RMB, respectively, and 7.4% of respondents’ annual household income was more than 300,000 RMB. Furthermore, the vast majority of respondents consumed milk every week, and specifically, the frequency of respondents’ milk consumption was “1–2 times” (30%), “3–4 times” (33.5%), and “5–6 times” (19.4%) per week. 56.1% of the respondents said that the proportion of imported dairy products accounted for less than 10% in their purchase of dairy products. In addition, the proportion of respondents who purchased imported dairy products from 10% to 30%, 30% to 50%, and more than 50% accounted for 21.3%, 12.9%, 9.7%, respectively.

When conducting the questionnaire survey, the concept of nutritionally enhanced food was first explained to all participants. That is, according to the needs of different groups of people, to maintain the original food nutritional ingredients or supplement the nutrients lacking in food, a certain amount of nutritional enhancements such as vitamins, minerals, and trace elements are added to the food to enhance its nutritional value [58,59,60]. Then, their cognition was investigated. As shown in Table 2, 41% of the respondents had never heard of nutritionally enhanced food, 31.9% had heard about but never purchased nutritionally enhanced food, and only 27.1% had heard about and purchased nutritionally enhanced food.

### 3.2. Model Estimation and Results

#### 3.2.1. Estimation Results of the RPL Model

We assumed that the coefficients of the “chooseno” and price were fixed and the parameters of the other imported milk attributes were randomly and normally distributed [61]. We use NLOGIT 5.0 to obtain the random parameter logit (RPL) model and latent class logit (LCL) model estimation results. As shown in Table 3, the coefficient of “chooseno” was negative and statistically significant, which means consumers can get higher utility from choosing any alternative to the non-buy option. In addition, the coefficient of price was negative and statistically significant, indicating that price increments decrease consumers’ utility. This is consistent with studies by Tempesta [21], Empen and Hamilton [62], and Yue et al. [63].

The coefficient of the skim attribute was positive and statistically significant at the 1% level, indicating that consumers prefer skim imported milk to whole imported milk. Chapman and Lawless [44], Harwood and Drake [45] found that American consumers would like skim milk rather than whole milk, but compared with skim milk, consumers preferred low-fat milk with 1% or 2% fat content. Yang et al. [64] reported that Chinese consumers prefer skim milk to 1.5% and 3.8% milkfat. Since our study did not set the level of low-fat, Chinese consumers’ preference for low-fat or skim milk needs further testing. 

The part-worth utility of the flavor attribute was negative and statistically significant at the 1% level, indicating that consumers highly valued this attribute and preferred plain imported milk. The coefficients of nutrition claims “contains vitamin A, vitamin D” and “contains Ca, Fe, Zn, and vitamin D” were positive and statistically significant at the 5% and 1% levels, respectively, indicating that consumers preferred milk with nutrition claims to milk without these claims. This is consistent with the research by Gulseven and Wohlgenant [59], who found that consumers prefer milk containing fiber and minerals to normal milk. Our research also showed that compared to imported milk with the claim “contains vitamin A, vitamin D,” consumers have a higher preference for imported milk with the nutrition claim “contains Ca, Fe, Zn, vitamin D.” The reason may be that consumers believe microelements are more difficult to obtain through the daily diet than vitamins. Furthermore, vitamin D facilitates calcium absorption, and simultaneously supplementing vitamin D and calcium benefits bone development and health, catering to the needs of families with children or the elderly.

Our results also show that for consumers, there is no significant difference between milk imported from New Zealand and that from France. In addition, the part-worth utility of the Germany level was positive and statistically significant at the 1% level, indicating that Chinese consumers preferred milk imported from Germany over milk from France and New Zealand. The reason may be that Germany is familiar to Chinese consumers because of its strict food safety control system and has gained their trust for many years. As a result, Germany surpassed New Zealand in 2013 to become the largest exporter of milk in China.

#### 3.2.2. Estimation Results of the LCL Model

A key issue in the latent class logit model is defining the number of consumer segments. The Akaike Likelihood Ratio Index should be calculated, which includes the Akaike Information Criterion (AIC); modified Akaike Information Criterion (AIC3); Bayesian Information Criterion (BIC); and ρ^2^. In general, the most suitable model has the lowest AIC, AIC3, and BIC, and the highest ρ^2^ [65]. As shown in Table 4, the improvement from three to four segments was greater than the change from four to five. Therefore, consumers are divided into four segments.

As shown in Table 3, in the LCL model for consumers in segment 1, the part-worth utilities of skim, Germany, claim “contains vitamin A, vitamin D” and claim “contains Ca, Fe, Zn, vitamin D” levels were positive. The part-worth utility of the flavor attribute was negative and significant at the 1% level. In addition, they had the strongest preference for imported milk with nutrition claims. Therefore, we called them “nutrition claim seekers,” who accounted for 57.3% of all participants. Consumers in segment 2 did not demonstrate a strong preference for nutrition claims, fat content, flavor, country of origin, and price attributes. The part-worth utilities of all attributes were not significant; therefore, those in this segment (6.6%) were called “indifferent” consumers. Although consumers in segment 3 preferred milk with the nutrition claim “contains vitamin A, vitamin D” and milk from New Zealand, they strongly preferred strawberry-flavored imported milk. Therefore, they were labeled “flavor-oriented” consumers, accounting for 14.6% of the total. Compared with the previous segments, consumers in segment 4 were extremely sensitive to price, and the part-worth utility of price attribute was negative and significant at the 1% level. Therefore, they were called “price-sensitive” consumers, accounting for 21.5% of all participants. “Price-sensitive” consumers also preferred skim milk, plain milk, and imported milk with the nutrition claim “contains Ca, Fe, Zn, vitamin D.”

#### 3.2.3. Economic Valuation Results

The average consumer WTP for the skim, New Zealand, and Germany attributes are 3.016 yuan/box, 0.281 yuan/box, and 2.831 yuan/box, respectively (Table 5). This means that Chinese consumers prefer skim milk rather than whole milk, and like milk imported from New Zealand and Germany more than from France. The results of this study also suggest that consumers have potential demands for the nutrition claims “contains vitamin A, vitamin D” and “contains Ca, Fe, Zn, vitamin D,” as the average WTP is 1.874 yuan/box and 3.812 yuan/box, respectively. This is consistent with Gulseven and Wohlgenant [66], who found that in the US, the value of milk increases by 2.5 cents per serving by adding vitamins and minerals to it. Francesco et al. [67] noted that in Italy, adding fiber to milk added a premium of 0.183 €/L, and a premium of 0.044 €/L and 0.343€/L could be charged if the product included added vitamins and calcium, respectively. In addition, the average consumer WTPs for strawberry and banana flavors are both negative, meaning that compared to strawberry-flavored and banana-flavored imported milk, consumers prefer plain milk and are willing to pay more to avoid strawberry- and banana-flavored imported milk.

In the LCL model, the preference and WTP for imported milk differed between the four segments. Among “nutrition claim seekers,” the WTP for the attribute claim “contains Ca, Fe, Zn, vitamin D” was more than 20 yuan/box. The premium price for skim was 18.292 yuan/box; 14.427 yuan/box for the claim “contains vitamin A, vitamin D”; 25.638 yuan/box for the claim “contains Ca, Fe, Zn, vitamin D”; and 16.888 yuan/box for Germany. Due to the frequently occurring food safety incidents in China, those who preferred some specific attributes of imported milk were willing to pay much more for them, which can explain why consumers’ WTPs were so high. In contrast, “indifferent” consumers had an extremely low WTP for each imported milk attribute. Their WTPs for all attributes were below 2.200 yuan/box. “Price-sensitive” consumers also had a relatively low WTP for each attribute. Their premium price for skim was 1.510 yuan/box; 2.459 yuan/box for the claim “contains Ca, Fe, Zn, vitamin D”; 0.661 yuan/box for New Zealand; and 1.891 yuan/box for Germany. “Nutrition claim seekers,” “indifferent,” and “price-sensitive” consumers all had negative WTPs for strawberry- and banana-flavored milk. Both “indifferent” and “price-sensitive” consumers had a negative WTP for the nutrition claim “contains vitamin A, vitamin D.” “Flavor-oriented” consumers’ WTP for the nutrition claim “contains Ca, Fe, Zn, vitamin D” was negative. Therefore, “nutrition claim seekers” valued nutrition claims more than other types of consumers. “Flavor-oriented” consumers paid more attention to milk flavor, and had the highest WTP for strawberry-flavored milk, which was 13.067 yuan/box. Their WTPs for the nutrition claim “contains vitamin A, vitamin D” and New Zealand attribute were 11.868 yuan/box and 9.498 yuan/box respectively. 

#### 3.2.4. Explaining Differences in Consumers’ Valuation

The results of the demographic characteristics of each class were reported in Table 2. As shown in Table 2 and Table 6, the majority of the 178 “nutrition claim seekers” were women, accounting for 58.4% of the total. Of these, 57.9% were aged 25–34 years, and 61.3% had a junior college or undergraduate-level education. The income level in this segment was moderate and most respondents’ (59.5%) annual household income ranged from 51,000 to 15,0000 RMB. These consumers consume milk at a higher frequency, and the proportion of those who drink milk every day (16.8%) accounts for the largest among the four types of consumers. The proportion of people who purchased imported dairy products as more than 50% in their dairy products purchase (12.4%) was the largest among the four groups of people. Their cognitive rate of nutritionally enhanced food was relatively high: 28.1% have heard about and purchased nutritionally enhanced foods, the largest proportion among the four segments of consumers. Consumers in this segment were most knowledgeable on the nutritional properties of food products, and used health and nutrition claims most frequently. Furthermore, they considered nutrition and health claims very important, scoring an average of 3.179 points or more on the 5-point scale. 

The vast majority of the 20 “indifferent” consumers were men, accounting for 80%. The proportion of consumers aged 18–24 years (45.0%) was the largest among the four types of consumers, and the frequency of weekly milk consumption was the lowest. The percentage of imported dairy products in their diary products purchasing was relatively low, and a vast majority of people (85.0%) purchased less than 30% imported dairy products in their dairy products purchasing. In general, “indifferent” consumers were less knowledgeable about nutritionally enhanced foods and nutrients. Among the four types of consumers, nutrition and health claims, excepting overall health claims, were the least important to them. In addition, these consumers were the least likely to use health and nutrition claims. This result supports Wardle et al. [68], Gulseven and Wohlgenant [66], and Huffman and Jensen [69], and this can be put down to the fact that men and young consumers have lower health awareness.

In contrast to “indifferent” consumers, the majority of the 45 “flavor-oriented” consumers were women, accounting for 71.1%. Most were aged 18–34 years. Compared with other types, “flavor-oriented” consumers had a lower education and income level, and their frequency of weekly milk consumption was moderate. The proportion of imported dairy products in their diary products purchasing was the lowest: 75.6% respondents purchased less than 10% imported dairy products in their dairy product purchasing and only 2.2% respondents purchased more than 50% imported dairy products. Although the proportion of consumers who have heard about nutritionally enhanced foods was the largest, the cognitive rates regarding carbohydrates, cholesterol, and other nutrients were relatively low. Health and nutrition claims were used significantly less frequently by “flavor-oriented” consumers than “nutrition claim seekers” and “price-sensitive” consumers. Compared to “nutrition claim seekers” and “price-sensitive” consumers, the nutrition and health claims were also less important to this group. 

There were 67 “price-sensitive” consumers, comprising a relatively equal share of female and male participants. Moreover, 73.2% were aged 25–44 years, and 59.6% had a junior college or undergraduate-level education. Of these, the annual household income of 19.4% ranged from 151,000–200,000, the largest proportion of the four segments. The percentage of people who purchased 30%-50% imported dairy products in their dairy products purchasing (22.4%) accounted for the largest among the four types of people. The consumers in segment 4 had the least knowledge about nutritionally enhanced food. The cognitive rate regarding nutrients and frequency of the use of nutrition and health claims were only lower than those for “nutrition claim seekers.” Nutrition and health claims were as important to this group as to “nutrition claim seekers.”

Overall, consumers would like overall health claims rather than nutrition claims, in line with the findings of Van Kleef et al. [70] and Williams et al. [71]. The reason may be that even if consumers are informed of nutrition claims, they often have difficulty understanding the specific role of nutrients, while health claims directly inform them of the role of some food or nutrients which is easier to understand.

## 4. Conclusions

In this study, 310 consumers in Shanghai participated in the investigation of consumer preference for imported milk attributes based on fat content, flavor, nutrition claim, country of origin, and price. People’s preferences and WTPs for imported milk attributes were assessed based on a choice experiment. We also explored consumers’ individual characteristics, cognition, attitudes, and purchasing habits, as they significantly differ in terms of choosing imported milk. The main conclusions are as follows:(a)The RPL results found that flavor is considered the most preferred attribute, followed by the nutrition claim, fat content, and country of origin attributes. For the flavor attribute, consumers prefer plain imported milk most, and are willing to pay additional prices to avoid strawberry- and banana-flavored imported milk. With regard to the fat content attribute, consumers like skim imported milk more. Furthermore, consumers are willing to pay an additional price for imported milk with the nutrition claim “contains vitamin A, vitamin D” and the nutrition claim “contains calcium, iron, zinc, vitamin D”. For the attribute of country of origin, consumers like Germany most, followed by New Zealand.(b)The results of the LCL model indicated that consumers in Shanghai have heterogeneous preferences for the consumption of imported milk. Based on consumers’ varying preferences for imported milk properties, they can be delineated into four segments: “nutrition claim seekers,” “indifferent,” “flavor-oriented,” and “price-sensitive” consumers. Among the four segments of consumers, “nutrition claim seekers” are more concerned about and have the highest WTPs for nutrition claims. The impact of imported milk attributes on “indifferent” consumers is minimal. “Flavor-oriented” consumers strongly prefer strawberry-flavored imported milk. “Price-sensitive” consumers concentrate on price so that they pay a relatively low price for each imported milk attribute.(c)Slightly more women than men are “nutrition claim seekers,” and most are aged 25–34 years. They have a moderate income level and a high frequency of milk consumption. For most “nutrition claim seekers,” imported dairy products plays an important role in their daily life. In addition, they are most knowledgeable about nutrients, and use nutrition and health claims most frequently. “Indifferent” consumers are mainly younger males who consume milk least frequently. They also have a lower cognition rate of nutrients and pay the least attention to nutrition and health claims when shopping. Most “flavor-oriented” consumers are women, and their cognition rate of nutrients and frequency of use of nutrition and health claims are lower than those of “nutrition claim seekers” and “price-sensitive” consumers. “Price-sensitive” consumers concentrate on price, and the prices paid for each attribute are low. Their use frequency of nutrition and health claims is only lower than “nutrition claim seekers.”

In accordance with the abovementioned conclusions, this paper has the following recommendations: (a)On the one hand, the Chinese government should strengthen public education and promote scientific publicity of nutritionally enhanced food and their functions, especially among young male consumer groups. This would reinforce and extend consumers’ understanding of nutrients and other attributes, guiding them to scientifically use nutrition and health claims and improve their ability of judgment regarding purchasing nutritionally enhanced milk. On the other hand, the Chinese government should strengthen supervision of the quality and safety of imported milk, ensure its good quality and favorable price, and accurately and timeously announce the quality and safety test results of imported milk to guide consumers’ rational demand for these products.(b)According to different consumer groups, milk production enterprises which focus on the Chinese market can formulate differentiated products to meet consumers’ needs and to sharpen the competitive edge of their own products. For example, for “nutrition claim seekers,” who are aged 25–34 years, have a moderate income level, a relatively high frequency of milk consumption, and prefer milk with nutrition claims, producers can produce milk that is skim, plain, contains vitamin A and vitamin D, or contains Ca, Fe, Zn, and vitamin D. For households with children or the elderly, producers can produce low-fat milk with a high calcium content. Finally, for consumers having a country of origin preference, Chinese overseas milk investment enterprises can select Germany and New Zealand as their milk production bases.

## 5. Limitations

The results are biased, because we considered a sample in Shanghai that is already familiar with the imported milk and has a relatively high recognition of nutrition claims. We didn’t consider other consumers who are not familiar with nutrition claims and imported milk. In the future, we should compare our results with the behavior of consumers who are not familiar with nutrition claims to see whether there is a difference.Due to the high cost, our research was biased on the choice experiment rather than the real choice experiment, which may overestimate consumers’ willingness to pay for imported milk.We designed the level of the price attribute in the choice experiment according to the real price of the normal imported milk on the Chinese market and from previous studies. We acknowledge that the range of price levels (4.5, 5.8, 7.1 and 8.4) may be too small to reflect the cost incurred in nutrition claims, fat content, country of origin. This may cause the WTP for attributes to be particularly high, not only for the nutrition claim attributes, but also for other attributes studied in this paper.The fourth limitation of our paper is that we allowed the respondents to evaluate their knowledge about nutrients on their own, which may not reflect their level of knowledge clearly.

## Figures and Tables

**Figure 1 ijerph-17-00244-f001:**
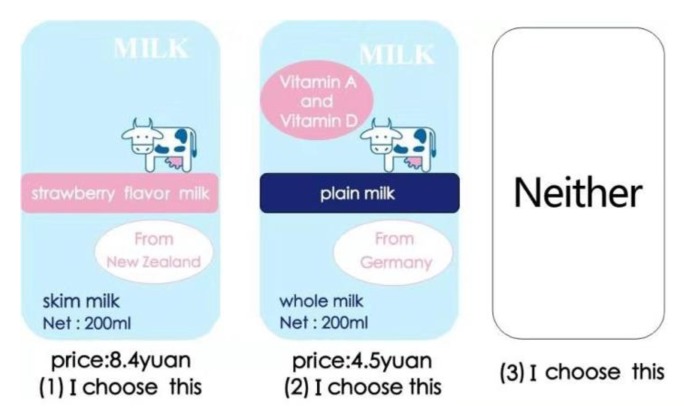
Sample choice task card.

**Table 1 ijerph-17-00244-t001:** Attributes and levels of imported milk.

Attributes	Levels
Fat content	Whole milk
	Skim milk
Flavor	Plain
	Strawberry flavor
	Banana flavor
Nutrition claim	Contains vitamin A, vitamin D
	Contains Ca, Fe, Zn, vitamin D
	No claim
Country of origin	New Zealand
	Germany
	France
Price	4.5 yuan/box
	5.8 yuan/box
	7.1 yuan/box
	8.4 yuan/box

**Table 2 ijerph-17-00244-t002:** Demographic characteristics and consumption habits.

Characteristics	Sample (n = 310)	Nutrition Claim Seekers (n = 178)	Indifferent (n = 20)	Flavor-Oriented (n = 45)	Price-Sensitive (n = 67)
**Gender** ***					
Male	43.9%	41.6%	80.0%	28.9%	49.3%
Female	56.1%	58.4%	20.0%	71.1%	50.7%
**Age** ***					
18–24	21.0%	18.5%	45.0%	26.7%	16.4%
25–34	54.8%	57.9%	30.0%	55.5%	53.7%
35–44	16.8%	18.0%	10.0%	11.1%	19.5%
≥45	7.4%	5.6%	15.0%	6.7%	10.4%
**Education Level**					
Primary school or lower	3.2%	2.2%	10.0%	4.4%	3.0%
Middle school	11.3%	12.4%	10.0%	17.8%	4.5%
High school	21.9%	18.5%	15.0%	31.1%	26.9%
Junior college	27.1%	27.0%	20.0%	24.5%	31.3%
Undergraduate	31%	34.3%	30.0%	22.2%	28.3%
Master or above	5.5%	5.6%	15.0%	0%	6.0%
**Annual household income (yuan)**					
50,000 yuan or less	14.2%	12.9%	5.0%	20.0%	16.4%
51,000–100,000 yuan	31.6%	34.8%	25.0%	22.2%	31.3%
101,000–150,000 yuan	24.5%	24.7%	35.0%	33.3%	14.9%
151,000–200,000 yuan	14.8%	14.0%	10.0%	13.4%	19.4%
201,000–300,000 yuan	7.4%	6.8%	15.0%	4.4%	9.0%
300,000 yuan or above	7.4%	6.8%	10.0%	6.7%	9.0%
**Weekly milk consumption** ***					
Never drink	4.2%	2.8%	15.0%	4.5%	4.5%
1–2 times	30%	28.1%	35.0%	33.3%	31.3%
3–4 times	33.5%	34.3%	30.0%	40.0%	28.4%
5–6 times	19.4%	18.0%	15.0%	20.0%	23.9%
Drink every day	12.9%	16.8%	5.0%	2.2%	11.9%
**Have you ever heard of or bought nutritionally enhanced food** ***					
I have heard of and purchased	27.1%	28.1%	20.0%	26.7%	26.9%
I have heard of but not purchased	31.9%	32.0%	35.0%	33.3%	29.8%
Never heard of	41.0%	39.9%	45.0%	40.0%	43.3%
**Percentage of imported dairy products in the purchase of dairy products** ***					
0–10%	56.1%	50.0%	65.0%	75.6%	56.7%
10–30%	21.3%	27.0%	20.0%	11.1%	13.4%
30–50%	12.9%	10.6%	5.0%	11.1%	22.4%
>50%	9.7%	12.4%	10.0%	2.2%	7.5%

Note: One-way ANOVA and post-hoc analysis were used to test the difference between segments. *** represents that the difference between segments is significant at 5% level.

**Table 3 ijerph-17-00244-t003:** Estimation results of RPL and LCL models.

Variables	RPL Model	LCL Model			
		Nutrition Claim Seekers	Indifferent	Flavor-Oriented	Price-Sensitive
Price	−0.1173 ***	−0.0123	−12.2229	−0.0659	−0.3067 ***
Skim	0.1768 ***	0.11268 ***	2.9184	−0.0248	0.2316 *
Strawberry flavor	−0.3494 ***	−0.16941 ***	−4.5124	0.4306 ***	−1.2052 ***
Banana flavor	−0.5220 ***	−0.1829 ***	−9.2510	−0.5080 ***	−1.5179 ***
Claim “contains vitamin A, vitamin D”	0.1099 **	0.0889 *	−2.7476	0.3911 ***	−0.0257
Claim “contains Ca, Fe, Zn, vitamin D”	0.2235 ***	0.1579 ***	13.3683	−0.1502	0.3771 **
New Zealand	0.0165	−0.0307	−4.1950	0.3130 **	0.1014
Germany	0.1660 ***	0.1040 **	7.0849	0.0067	0.2900
Chooseno	−1.3381 ***	−2.0426 ***	−93.7504	0.3928	−1.2880 **
Standard deviation estimation					
Skim	0.2953 ***	−	−	−	−
Strawberry flavor	0.7066 ***	−	−	−	−
Banana flavor	0.6153 ***	−	−	−	−
Claim “contains vitamin A, vitamin D”	0.3656 ***	−	−	−	−
Claim “contains Ca, Fe, Zn, vitamin D”	0.3912 ***	−	−	−	−
New Zealand	0.2194 *	−	−	−	−
Germany	0.2569 **	−	−	−	−
Class Prob.	NA	0.5734	0.0655	0.1462	0.2149
Number of observations	3410	1958	220	495	737
Pseudo R-squared	0.1268	0.1727			
Log-likelihood	−3271.0879	−3099.3374			
		3410			

Note: ***, **, and * denote significance at the 1%, 5%, and 10% significance levels, respectively.

**Table 4 ijerph-17-00244-t004:** Statistics to determine the optimal number of consumer segments.

Segments	Parameters (P)	Log Likelihood (LL)	AIC	AIC3	BIC	ρ^2^
2	19	−3184.81366	6407.62732	6426.62732	3262.0911	0.14480
3	29	−3136.65355	6331.30710	6360.30710	3254.6033	0.15498
4	39	−3099.33742	6276.67484	6315.67484	3257.9595	0.15707
5	49	−3100.68036	6299.36072	6348.36072	3299.9748	0.15925

Note: AIC = –2(LL – P); AIC3 = (–2LL + 3P); BIC = (–LL + (P/2) × ln(N)); ρ^2^ = (1 – AIC/2Restricted LL); Restricted Log-likelihood = –3746.26790.

**Table 5 ijerph-17-00244-t005:** Consumer WTP for each level (yuan/200 mL).

Variables	The Sample	Nutrition Claim Seekers	Indifferent	Flavor-Oriented	Price-Sensitive
**Skim**	3.016 [1.936,4.096]	18.292 [9.063,27.521]	0.478 [−0.364,1.319]	−0.753 [−7.741,6.262]	1.510 [−0.140,3.161]
**Strawberry flavor**	−5.959 [−7.727,−4.190]	−27.501 [−41.364,−13.641]	−0.738 [−2.078,0.602]	13.067 [4.245,21.889]	−7.859 [−10.635,−5.084]
**Banana flavor**	−8.902 [−10.691,−7.114]	−29.685 [−45.924,−13.446]	−1.514 [−4.095,1.067]	−15.414 [−26.349,−4.480]	−9.898 [−12.869,−6.928]
**Claim “contains vitamin A, vitamin D”**	1.874 [0.306,3.441]	14.427 [−0.875,29.729]	−0.450 [−1.318,0.419]	11.868 [3.600,20.137]	−0.168 [−2.543,2.207]
**Claim “contains Ca, Fe, Zn, vitamin D”**	3.812 [2.299,5.325]	25.638 [11.253,40.021]	2.187 [−1.580,5.955]	−4.557 [−13.565,4.451]	2.459 [0.098,4.821]
**New Zealand**	0.281 [−1.134,1.695]	−4.989 [−19.041,9.063]	−0.686 [−1.734,0.361]	9.498 [0.370,18.626]	0.661 [−1.707,3.029]
**Germany**	2.831 [1.378,4.284]	16.888 [3.044,30.732]	1.159 [−1.058,3.376]	0.203 [−8.843,9.249]	1.891 [−0.629,4.412]

Note: Values in the brackets are 95% confidence interval.

**Table 6 ijerph-17-00244-t006:** Knowledge, attitude, and use of nutrition and health claims.

	Sample	Nutrition Claim Seekers	Indifferent	Flavor-Oriented	Price-Sensitive	Sig.
Knowledge on carbohydrates^a^	77.7%	80.2%	76.2%	65.2%	80.3%	0.000
Knowledge on fibre^a^	78.4%	81.4%	61.9%	74.0%	78.8%	0.000
Knowledge on cholesterol^a^	78.7%	83.1%	81.0%	60.9%	78.8%	0.000
Importance of claims on fat^b^	3.175	3.237	2.857	3.064	3.182	0.000
Importance of claims on sugars^b^	3.196	3.209	2.667	3.175	3.333	0.000
Importance of claims on vitamins^b^	3.404	3.457	2.952	3.306	3.470	0.000
Importance of health claims in general^b^	4.010	4.079	4.095	3.827	3.924	0.000
Importance of claims on “Omega-3 fatty acids help to maintain heart health”^b^	3.495	3.536	2.905	3.456	3.606	0.000
Importance of claims on “Zinc and iron contributes to normal cognitive brain function”^b^	3.778	3.831	3.238	3.782	3.803	0.000
I often use health claims on food while food shopping^b^	3.050	3.203	2.381	2.868	2.970	0.000
I often use nutrition claims on food while food shopping^b^	3.071	3.179	2.619	2.825	3.106	0.000

Note: ^a^ A survey of variables related to the perceptions of nutrients such as carbohydrates, level of knowledge on nutritional properties (uninformed = 0, informed = 1); ^b^ Variables on consumer attitudes, interests, and habits of nutritional information were measured using a five-point scale (e.g.,: “not important,” “less important,” “general,” “more important,” and “very important” were indicated as 1–5 points, respectively). ^c^ Significant differences in cognition, attitude, and habits among the four types of consumers were estimated in an ANOVA analysis. Sig. < 0.05 indicates significant differences among the four groups.

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
