# Peer review of "Consumers’ Willingness to Pay for Imported Milk: Based on Shanghai, China"

_ijerph, 2019, doi:10.3390/ijerph17010244_

Round 1

Reviewer 1 Report

ROUND 1

Brief summary

This manuscript is very interesting with a strong link between a problem of public health, related to an important feed intake, and public and private management. It is suitable to publish in International Journal of Environmental Research and Public Health in order to the aim and scope of this review. Its main contributions could support decision-making processes of health policy makers as well as of managers of state or private enterprises. The model is well implemented following the comprehensive recommendations to develop a choice experiment of a contingent valuation method. It is serious and rigorous but authors would improve one important aspect. This paper contains a large amount of numerical data and many of them are completely unnecessary. This problem is recurrent in all section and particularly problematic in abstract, introduction and conclusion sections. These sections would be more qualitative and don´t reproduce information of results section over and over again.

Broad comments

Abstract section: This part should only contain the purpose of the investigation, the way to achieve this and the most important findings. And these items should be always submitted to readers in a qualitative manner

Introduction section: It is not all clear what the problems are. It is also no clear what the hypothesis to contrast are. Apparently lines 120 to 129 of literature review section would be the starting hypothesis. But authors reach this milestone after a lengthy process of three pages. I recommend merge sections 1 and 2 and compose and introduction section with more clarity and simplicity. On the other hand, it is not very common to insert a figure in an introduction section, if it is not absolutely necessary. And figure 1 does not seem the case.

Methods section:

In line 133 authors insert reference number 30. This investigation refers to valuation of public goods in general and environment goods in particular. Authors would discuss about the suitable to adapt this methodology to the study object based on previous studies Authors should discuss the attributes (section 3.1.1) in the stated order in line 141 to 143. Why price attribute appear at the end of this section? The analysis of “country of origin” attribute is poor in relation to the rest of attributes. Authors could use related information of introduction not suitable for this section

Results sections:

The section 4.2.1 should be relocated in Methods section Authors should no repeat in the text numerical data that they already are in tables. This also makes it difficult for readers to understand the main aspects of the discussion of the results In lines 382 to 385 and 395 to 396 the estimated WTP for that attributes are extremely high and authors don´t discuss about this. This results draw the attention. Why authors don’t discuss about this findings? What are the causes?

Discussion section:

This section should clearly name Conclusion section Please remove the majority of the data to draw conclusions in a more qualitative manner Why authors don’t include limitations to the investigation?

Specific comments

In line 38 and footnote 1 authors refer to Mintel´s report but this information doesn´t appear in reference section In lines 32 and 33 authors said “most imported milk in China is sold through online shopping platforms, further reducing the circulation cost”: citation needed Consequently authors said “As such, the price of imported milk has decreased by 24.7% from $1800/t in 2010 to 34 $1355/t in 2018”. This is a strong conclusion that derives of above statement and authors should discuss about this or to remove it. In line 133 authors insert reference number 31. This reference does no clarify why authors select the five following attributes. Perhaps only one of them In line145 and following authors refer to sugar. Why? This creates confusion

Reviewer 2 Report

I think it's a good article. I think the paper has a lot of virtues. However, in order to improve the final product, I make some recommendations.

1) It is not a good idea to include keywords in the article title. Then, I recommend changing "imported milk" and "willingness to pay" as keywords.

2) The first time you use an acronym, you must write its meaning. Then, you need to type, at least once, the full name of LCL and RPL.

3) What software do you use to obtain the LCL?

4) Lines 380, 381 and 382 (page 11) are redundant with 383, 384 and 385. I recommend removing lines 380, 381 and 382.

4) On line 405 (page 12), there is an error, because the household income range should be 51,000 to 150,000.

5) The account of men's percentage on line 415 (page 12) should be an error because 81% of men do not match 80.0% of table 2.

6) I believe that ANOVA is not sufficient to know the differences between the four consumer groups. ANOVA says that there are differences, but not between which specific groups. I recommend doing a Post-Hoc analysis.

7) Likert scales have the answer options "completely agree", "agree", etc., but not "not important", "less important", etc., as a semantic differential scale.

8) There is a typo on line 467 (page 14) with the word "arrribute".

9) The discussion is not right because you have repeated the results. I recommend that some comments you wrote under the heading of results (epigraph 4) pass to the discussion epigraph.

Best regards,

Round 2

Reviewer 1 Report

Authors have improved the manuscript following the recommendations suggested in round 1.

I recommend accept this paper in the present form

Reviewer 2 Report

In my opinion, the manuscript has been significantly improved and now guarantees publication in IJERPH after minor revision of text editing.